# Initial Data and a Clinical Diagnosis Transition for the Aiginition Longitudinal Biomarker Investigation of Neurodegeneration (ALBION) Study

**DOI:** 10.3390/medicina58091179

**Published:** 2022-08-30

**Authors:** Nikolaos Scarmeas, Argyro Daskalaki, Faidra Kalligerou, Eva Ntanasi, Eirini Mamalaki, Antonios N. Gargalionis, Kostas Patas, Stylianos Chatzipanagiotou, Mary Yannakoulia, Vasilios C. Constantinides

**Affiliations:** 11st Department of Neurology, Aiginition Hospital, National and Kapodistrian University of Athens, Medical School, 11528 Athens, Greece; 2Taub Institute for Research in Alzheimer’s Disease and the Aging Brain, The Gertrude H. Sergievsky Center, Department of Neurology, Columbia University, New York, NY 10032, USA; 3Department of Nutrition and Diatetics, Harokopio University, 17671 Athens, Greece; 4Department of Medical Biopathology, Aiginition Hospital, 11528 Athens, Greece

**Keywords:** Alzheimer’s disease, biomarker, CSF biomarker, neurodegeneration, ALBION study

## Abstract

*Background and Objectives: *This article presents data from the ongoing Aiginition Longitudinal Biomarker Investigation of Neurodegeneration study (ALBION) regarding baseline clinical characterizations and CSF biomarker profiles, as well as preliminary longitudinal data on clinical progression. *Materials and Methods: *As of March 2022, 138 participants who either were cognitively normal (CN, *n* = 99) or had a diagnosis of mild cognitive impairment (MCI, *n* = 39) had been recruited at the specialist cognitive disorders outpatient clinic at Aiginition Hospital. Clinical characteristics at baseline were provided. These patients were followed annually to determine progression from CN to MCI or even dementia. CSF biomarker data (amyloid β1-42, phosphorylated tau at threonine 181, and total tau) collected using automated *Elecsys*^®^ assays (*Roche* Diagnostics) were available for 74 patients. These patients were further sorted based on the AT(N) classification model, as determined by CSF Aβ42 (A), CSF pTau (T), and CSF tTau (N). *Results:* Of the 49 CN patients with CSF biomarker data, 21 (43%) were classified as exhibiting “Alzheimer’s pathologic change” (A+Τ– (Ν)−) and 6 (12%) as having “Alzheimer’s disease” (A+T–(N)+, A+T+(N)–, or A+T+(N)+). Of the 25 MCI patients, 8 (32%) displayed “Alzheimer’s pathologic change”, and 6 (24%) had “Alzheimer’s disease”. A total of 66 individuals had a mean follow-up of 2.1 years (SD = 0.9, min = 0.8, max = 3.9), and 15 of those individuals (22%) showed a clinical progression (defined as a worsening clinical classification, i.e., from CN to MCI or dementia or from MCI to dementia). Overall, participants with the “AD continuum” AT(N) biomarker profile (i.e., A+T–(N)–, A+T–(N)+, A+T+(N)–, and A+T+(N)+) were more likely to clinically progress (*p *= 0.04). *Conclusions: *A CSF “AD continuum” AT(N) biomarker profile is associated with an increased risk of future clinical decline in CN or MCI subjects.

## 1. Introduction

Based on the 2022 Alzheimer’s disease (AD) Facts and Figures [1], the number of people in the U.S. aged 65 and older with Alzheimer’s dementia is projected to reach 12.7 million by 2050; the same is true for nearly 18.8 million people in the wider European region [2]. Estimated prevalence rates are based on clinical diagnoses. 

The neuropathological hallmarks of AD are amyloid plaques and neurofibrillary tangles. In the early 1980s, the “amyloid hypothesis” was introduced [3]. This hypothesis stated that the extracellular accumulation of pathologically misfolded amyloid β (Aβ) species (Aβ with 42 amino acids in particular) in the form of amyloid plaques was likely the primary event driving AD pathogenesis. Amyloid pathology can be established in vivo, either by a decrease in the CSF β-amyloid (Aβ42) concentration or by an increase in amyloid-specific radiotracer uptake, as measured by PET CT. On the other hand, the “tau hypothesis” proposed a mechanism of neurotoxicity based on the loss of function of microtubule-stabilizing tau proteins caused by hyper-phosphorylation, which results in the degradation of the cytoskeleton [4]. Tau pathology can be established by an increase in CSF phosphorylated tau species, such as phosphorylated tau in threonine 181 (pTau). An increase in the level of total CSF tau proteins (tTau), on the other hand, is a non-specific marker of neurodegeneration. 

A CSF biomarker profile of increased levels of pTau and total tTau, in addition to a decrease in Aβ42, is highly diagnostic of an underlying AD pathology in vivo and predates cognitive decline by many years. This CSF AD profile is highly predictive of the progression of cognitively unimpaired subjects to mild cognitive impairment (MCI) and dementia [5]. 

Advances in the field of neurochemical (CSF and plasma) and PET CT biomarkers have shifted the theoretical framework of the definition of Alzheimer’s disease from a clinical to a biological entity. This shift in the definition of AD was promoted in the most recent Research Framework for AD [6] by the National Institute on Aging and the Alzheimer’s Association (NIA-AA). In short, this research framework categorizes biomarkers of different modalities into three categories: amyloid pathology (A), tau pathology (T), and neurodegeneration (N). Based on this AT(N) classification system, three major “biomarker categories” are defined: (1) normal AD biomarkers, (2) the Alzheimer’s continuum (either Alzheimer’s pathologic change or Alzheimer’s disease), and (3) non-AD pathologic change. Thus, AD is considered a biological–clinical continuum with a biomarker-supported preclinical stage that progresses to MCI and then to dementia due to AD. The IWG-2 diagnostic criteria also recommend using either CSF biomarkers or PET imaging for the classification of patients with typical amnestic AD, as well as for those with atypical and mixed forms of AD [7]. 

Importantly, neuropathological changes in AD may precede clinical manifestations by many years. During this preclinical AD phase, amyloid-related biomarkers seem to precede neurodegeneration biomarkers [8]. The preclinical stage of AD is of pivotal importance for the development of disease-modifying AD treatments to be delivered before the advancement of irreversible neurodegeneration. To this extent, longitudinal studies of AD biomarkers in cognitively unimpaired subjects are paramount. 

Herein, we present some initial findings of the Aiginition Longitudinal Biomarker Investigation of Neurodegeneration study (ALBION) regarding the implementation of the AT(N) classification system based on CSF AD biomarkers in a cohort of cognitively normal and MCI patients, as well as preliminary data regarding clinical progression. Overall, the aim of the ALBION study is to answer various research questions about aging, subjective cognitive decline, and the preclinical and precursor stages of the most common cause of dementia (i.e., Alzheimer’s disease), as well as to explore possible biomarkers for its early diagnosis, prediction, and primary prevention.

## 2. Materials and Methods

### 2.1. Participants

The ALBION protocol was approved by the Institutional Review Board of the Aiginition University Hospital of Athens. All participants provide written informed consent at the time of enrollment. All participants are individuals aged 40 years or older who are either referred by other specialists or self-referred to the cognitive disorders outpatient clinic of a tertiary university hospital. These individuals may have subjective memory complaints or a positive family history, or they may be asymptomatic with a commitment to contributing to medical science. A clinical diagnosis is established by a specialist neurologist after an extensive standardized neuropsychological assessment. Only those considered to be cognitively normal (CN) or who have MCI, based on established diagnostic criteria [9,10], are included for the purposes of this study. An MCI diagnosis is assigned when the participant has cognitive complaints and a measurable deficit in cognition with a standard deviation (SD) below 1.5 in at least one domain, in the absence of dementia or impairment in everyday functioning. All patients with a diagnosis of dementia [11,12,13] were not enrolled, i.e., were excluded from this study, as were patients with neurological, psychiatric, or medical conditions associated with a high risk of cognitive impairment.

### 2.2. Methods

A detailed description of the design of the ALBION study has been published elsewhere [14]. Herein, we provide a snapshot of the baseline assessment (Figure 1: First assessment) in order to briefly summarize the variety of bio-samples and biomarkers that are being collected and will be available for further analysis.

With respect to magnetic resonance imaging, all participants without obvious medical contraindications are encouraged to undergo whole-brain imaging on a 3T Philips Achieva-Tx MR scanner (Philips, Best, The Netherlands) equipped with an eight-channel head coil. The imaging protocol contains basic sequences used in clinical practice and research, and a formal report is provided by a neuroradiologist.

A thorough interview and a clinical examination are performed by neurologists at baseline. Vital signs, anthropometric measurements, physical strength data (using a handgrip dynamometer), and physical performance data (such as 4-meter gait speed) are recorded. Furthermore, participants are kindly invited to provide information in a range of questionnaires related to neuropsychological and lifestyle issues, such as a subjective cognitive complaints questionnaire [15,16], the hospital anxiety and depression scale (HANDS) [17], a leisure time activities questionnaire [18], a sleep questionnaire [19], and a questionnaire on physical activity [20].

With respect to cognitive assessments, a full cognitive evaluation is administered by trained neuropsychologists. Global cognition is assessed using the Mini Mental State Examination (MMSE) [21] and the Addenbrooke’s Cognitive Examination-revised (ACE-r) [22], while the pre-morbid level is estimated based on vocabulary. Data from a variety of neuropsychological tests are used to provide information for five main cognitive domains: (a) attention (Trail Making Test A [23] and Digit Span Forwards [24]), (b) executive function (Trail Making Test B [23], the Stroop Test [25], and Digit Span Backwards [24]), (c) visuo-spatial (the Medical College of Georgia Complex Figure Test/copy and the visuo-spatial component of ACE-r), (d) memory (verbal memory: the Greek Verbal Learning Test and story recall, both immediate and delayed [26]; nonverbal memory: the Medical College of Georgia Complex Figure Test, both immediate and delayed), and (e) language (the semantic and phonological verbal fluency component of ACE-r, the language component of ACE-r, and a 40-item naming test).

For an assessment of clinical status, the clinical diagnosis is determined independently and blinded with respect to the CSF biomarker report for each participant. An experienced neurologist in the cognitive disorder field evaluates all information provided by the individual, as well as any reliable collateral history, in conjunction with the objective scores from the comprehensive battery of neuropsychological tests.

With respect to cerebrospinal fluid (CSF) biomarkers, the lumbar puncture (LP) procedures, as well as the collection, processing, and storage of the CSF, are conducted according to international guidelines [27]. To date, the majority of the samples have been analyzed for Aβ42, pTau, and tTau either using automated *Elecsys*^®^ assays (*Roche* Diagnostics) or by ELISA (Innotest, Fujirebio Europe). In previous studies, AD CSF biomarkers analyzed by Elecsys showed a high concordance with Amyloid-β PET, which is considered a reliable alternative whenever nuclear medicine is not available [28,29]. For the classification according to the AT(N) system in the current study, we considered CSF Aβ42 for A, CSF pTau for T, and tTau for (N). The provided reference ranges for a positive result were as follows: Aβ42 ≤ 1000 pg/mL, pTau > 27 pg/mL, and tTau > 300 pg/mL. Accordingly, three main biomarker categories emerged: (1) normal AD biomarkers (A–-T–(N)–), (2) AD continuum (either AD-pathologic change, i.e., A+T–(N)–, or AD disease, i.e., A+T–(N)+, A+T+(N)–, or A+T+(N)+), and (3) non-AD-pathologic change (A–T–(N)+, A–T+(N)–, or A–T+(N)+). Our goal is to be able to process all the available samples by automated *Elecsys*^®^ in the near future.

Blood sample collection is performed on the date of the LP for the first visit and then is repeated every two years. Serum, plasma, buffy coat, and RNA samples are appropriately stored at −80 degrees C for future analysis.

All participants are referred for an electroencephalogram (EEG) at their first and third visits. EEG recordings are acquired with a 10/20 international system of electrode placement method. A 15-min-awake resting-state EEG recording is obtained with a Micromed SAM 25FO 32-channel headbox (Micromed, Treviso, Italy) at a sampling rate of 1024 Hz (analog high-pass filter 0.15 Hz) with scalp gold electrodes. Also processed are a spectral analysis of the posterior dominant rhythm (PDR) for 3 s after each eye closure and a novel non-linear measure of eyes-open versus eyes-closed EEG synchronization.

A wrist Actiwatch 2 (Phillips-Respironics) [30] is used to record sleep parameters for a 7-day period along with a WatchPAT [31], which could be considered a clinically reliable tool for diagnosing obstructive sleep apnea/OSA. The provided parameters, in association with the subjective sleep questionnaire tool, enable us to review the impact of circadian and sleep disturbances in neurodegenerative diseases.

For dietary intake, four 24-h dietary recalls are collected; three are collected by telephone, and one on-site on the date of the LP procedure. Trained dieticians ask the participants to report, in detail, all foods and beverages consumed during the previous day. Data are analyzed in terms of nutrients using the dietary analysis software Nutritionist Pro™ (2007, Axxya Systems, San Antonio, TX, USA).

In conjunction with the information collected about dietary habits, participants are also prompted to provide a stool sample, kept in the right storage conditions until delivery, in order to conduct gut microbiota testing [32]. The role of the gut microbiota in the development of neurodegenerative diseases such as AD has attracted a lot of interest from researchers lately [33]. Beneficial modifications to the gut microbiota could reveal new recommendations and therapeutic approaches.

### 2.3. Statistical Analysis

Descriptive analyses were performed using IBM SPSS version 20 (IBM Software Inc., Hong Kong, China). Data are presented initially according to baseline clinical status and further according to AT(N) biomarker profile, wherever such information was available. Continuous variables are expressed as mean values (SD), while categorical variables are referred to as frequencies and percentages. To compare continuous variables, we used either an independent samples t-test or an analysis of variance. For nominal variables, we used either the chi-square test or the Fisher exact test. ApoE genotyping was dichotomized into ε4 carriers and non-carriers.

We conducted additional analyses, collapsing the AT(N) biomarker profile into fewer categories: we classified the subcohort with available biomarker data into an “AD continuum” group and a “non-AD continuum” group (i.e., those with either “normal AD biomarkers” or “non-AD pathologic change”). In order to explore the association of clinical progression with the “AD continuum” profile, we dichotomized the participants into progressors (either to MCI or dementia for CN participants or to dementia for MCI participants) and non-progressors (either remaining stable or even MCI participants converting to CN), and applied the chi-square test. The statistical significance limit was set at a *p*-value < 0.05.

## 3. Results

A total of 138 individuals had been recruited as of March 2022, and their main characteristics are presented in Table 1. There is a female predominance in this study cohort. More than two-thirds (71.7%) were classified as cognitively normal at the baseline visit. Overall, almost half had a positive family history of dementia (either paternal or maternal or both). As expected, the MCI group had lower MMSE scores. ApoE genotyping is provided where available, i.e., for 114 of the 138 subjects, with a 27.2% overall frequency of ε4 carriers. The proportion of ε4 carriers appears higher in the MCI group.

A CSF biomarker report was available for 74/138 participants. Comparing these 74 individuals to the remaining 64 with no CSF biomarker profile yet provided (Table 2), no significant differences were observed in sex, education or MMSE scores, or baseline clinical diagnosis (all *p* > 0.05) Therefore, we may assume that the subgroup with available CSF biomarkers is more or less representative of the overall study sample.

From the subgroup with available CSF biomarkers, 49/74 subjects (66%) were classified as CN subjects at baseline. Data regarding the AT(N) model are presented in Table 3. There were no significant differences in sex, education, or MMSE among AT(N) categories. ApoE ε4 varied by AT(N) category, with the “AD continuum” group containing the highest number of carriers (12/20 individuals, *p* = 0.005). The AT(N) classification did not vary by diagnostic group. Classification in the “AD continuum” (vs. the ”non-AD continuum”) biomarker category was similar for the CN (27 of 49 subjects) and MCI (14 of 25 subjects) subgroups, with *p *= 0.94.

Of the 49 CN participants, 45 had further evaluations (min = 2, max = 5) within a mean follow-up period of 2.2 years (SD = 0.94, min 0.83, max = 3.92) (Table 4). No CN participants with baseline normal AD biomarkers progressed clinically to either MCI or dementia. Only CN participants with a baseline “AD disease” AT(N) profile progressed to clinical dementia. The respective frequencies for MCI conversion were 5% for the “AD pathologic change” group and 33.3% for the “non-AD pathologic change” group. Five participants who were initially sorted into the “AD continuum” group showed further clinical progression, but that was not statistically significant compared to the one individual from the “non-AD continuum” group who had also progressed as of the latest diagnosis (*p* = 0.222*).*

Table 5 shows the progression of 21 of the 25 MCI individuals, who had a minimum of two and a maximum of four assessments during a mean follow-up of 1.87 years (SD = 0.87, min = 0.83, max = 3.42). Nine of the twenty-one MCI participants (42.9%) remained stable as per clinical diagnosis, while nine participants (42.9%) progressed to dementia. Interestingly, three individuals (13.6%) reverted to a CN state as per the final diagnosis. These three participants had either “normal AD biomarkers” at baseline (*n* = 2) or an “AD pathologic change” profile (*n* = 1). Progression to dementia seemed more frequent in the “AD continuum” biomarker group (seven of nine), but that was not statistically significant compared to the “non-AD continuum” progressors (*p* = 0.184*).*

## 4. Discussion

In the current study, we provide preliminary data on the clinical course of subjects with regard to their CSF biomarker profiles. No difference was observed regarding the basic clinical characteristics at baseline, apart from the fact that the proportion of ApoE carriers was higher within the AD continuum group. In our cohort, subjects with an AD continuum CSF profile were more likely to deteriorate cognitively over time.

These provisional results agree with the literature [34,35,36]. Moreover, the majority of the AD pathologic change group, which was initially defined as CN, retained its fair cognitive performance through the years, providing reassurance that the early stage of the AD continuum is likely of long duration. On the other hand, it raises questions regarding how the MCI individuals who reverted to normal will fare in the future. Treating depressive symptoms, along with the learning effect, probably affected the diagnoses in the latest assessment. Similarly, the question of whether to disclose AD-relevant biomarker results in research settings has been debated for years [37]. Thus, information regarding clinical progression must always be interpreted with caution, especially when a study population is concerned. Data regarding the etiology of the final outcome other than AD dementia will be presented in the future.

In the absence of disease-modifying treatments, primary prevention remains the main strategy for reducing AD prevalence and incidence [38]. According to the latest AD development drug pipeline [39], there are 143 agents in 172 trials of treatments for AD, with 31 agents in 47 Phase 3 trials. Twenty-one (67.8%) agents in Phase 3 trials are disease-modifying treatments (DMTs), but only six of them (29%) project amyloid as the main mechanism of action. Finally, 6 out of 47 trials in Phase 3 are prevention trials that enrolled, by definition, cognitively normal participants known to be at risk for AD (i.e., at the preclinical AD stage).

Over the past two decades, observational studies recruiting cognitively healthy subjects have focused on normal aging and its relationship to cognitive decline and dementia [40,41,42]. These studies have provided insight into the complex interplay between clinical features, cognitive performance, risk factors (either modifiable or fixed), and variable biomarkers in well-characterized populations.

Additionally, many studies have focused specifically on cognitively intact patients at high risk of developing dementia due to AD [43,44,45,46]. The main aim of these studies is the identification of biomarkers that can accurately predict AD pathology in vivo, as well as provide information on the temporal relationship between amyloid, tau, and neurodegeneration biomarkers and cognitive/functional status. Moreover, many studies have focused on the effect of modifiable risk factors on AD pathological processes [47,48].

We recognize that our study has some limitations that should be mentioned. ALBION, by definition, recruits from a specialist cognitive disorders outpatient clinic, and half of the participants had a family history of late-onset AD dementia. All participants were Caucasian. Consequently, referral bias exists, and our results may have relatively low generalizability. In addition, the relatively short length of follow-up may have diminished our capacity to reveal substantial changes in the cognitive and clinical status of our participants. Thus, a longer duration of follow-up is essential. This is even more pertinent in our cohort that recruits from a relatively younger age. Moreover, the group exhibiting non-AD-pathologic change was quite small (only *n* = 3), limiting our ability to adequately capture the biomarker heterogeneity of these patients. Overall, the sample size of patients with available further follow-up assessments was too small; that fact could easily increase the likelihood of Type II error, limiting the power of the study. Finally, CSF biomarker assessment has not yet been completed for all participants; however, our CSF subcohort seemed quite similar to the whole study population.

The ALBION study has some strengths to be noted. The main aim of the study is to systematically collect clinical and paraclinical data on subjects in the preclinical stages of AD beyond a single moment in time. To this extent, a multitude of detailed clinical information, as well as very extensive neuropsychological, neurophysiological, reliable imaging, and biomarker data have been collected. Additionally, the assessments are performed quite frequently, at annual intervals. This holistic approach, which takes place in a specialist clinic of a tertiary university hospital, may facilitate the investigation of the mechanisms that contribute to the pathogenesis of cognitive disorders and modify their clinical course. We use an established method for CSF biomarker assessments. Our study continues to collect longitudinal data (so far, up to 3.5 years) that allow estimations of clinical progression over time based on variations within an individual. Clinical classifications are carried out by clinicians with subspecialty training and considerable experience in the cognitive disorders field. They are also blind to CSF biomarker status.

## 5. Conclusions

Overall, our preliminary data highlight the importance of CSF biomarkers in determining cognitive decline in cognitively unimpaired subjects and individuals with MCI. They also demonstrate, once more, the underlying heterogeneity of patients, both in terms of CSF biomarker profile but also in terms of future clinical course.

## Figures and Tables

**Figure 1 medicina-58-01179-f001:**
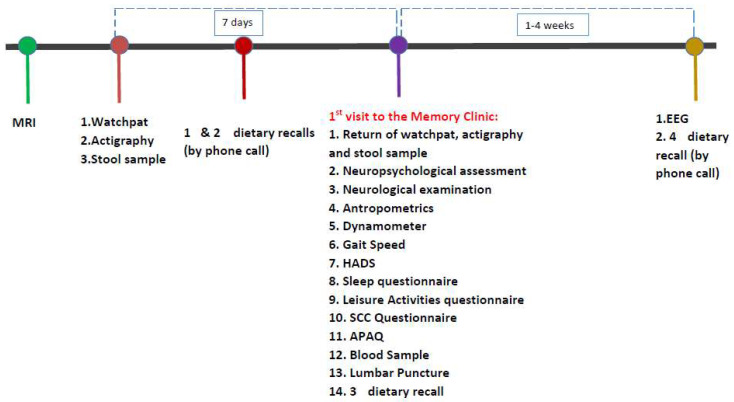
First assessment.

**Table 1 medicina-58-01179-t001:** Participants’ characteristics at baseline by clinical diagnosis.

Variable	CN (*n* = 99)	MCI (*n* = 39)	All (*n* = 138)	*p*-Value
Sex, female (%)	67 (67.7)	24 (61.5)	91 (65.9)	0.493
Age, y, mean (SD)	62.9 (9.2)	65.9 (8.2)	63.8 (9.0)	0.089
Education, y, mean (SD)	13.6 (3.7)	12.5 (4.1)	13.3 (3.8)	0.144
Family history of dementia, *n* (%)	45 (45.5)	18 (46.2)	63 (45.7)	0.941
MMSE, mean (SD)	28.8 (1.4)	26.5 (2.0)	28.2 (1.9)	<0.001
ApoE ε4 carrier, n/N performed (%)	18/81 (22.2)	13/33 (33.3)	31/114 (27.2)	0.062

Abbreviations: CN = cognitively normal, MCI = mild cognitive impairment, SD = standard deviation, MMSE = Mini Mental State Examination.

**Table 2 medicina-58-01179-t002:** Participant characteristics for those with and without CSF biomarker profiles.

	Participants with Biomarkers (*n* = 74)	Participants without Biomarkers (*n* = 64)	*p*-Value
Sex, female (%)	47 (34.1)	44 (31.9)	0.517
Age, y, mean (SD)	64.5 (9.0)	63.0 (9.0)	0.329
Education, y, mean (SD)	13.0 (3.7)	13.6 (3.9)	0.332
MMSE, mean (SD)	28.13 (1.81)	28.25 (2.1)	0.717
Clinical diagnosis			
CN	49	50	0.121
MCI	25	14	

Abbreviations: CSF = cerebrospinal fluid, SD = standard deviation, MMSE = Mini Mental State Examination, CN = cognitively normal, MCI = mild cognitive impairment.

**Table 3 medicina-58-01179-t003:** Participant characteristics at baseline by AT(N) biomarker categories.

	Normal AD Biomarkers	AD Continuum		Non-AD Pathologic Change	*p*-Value
		AD Pathologic Change	AD Disease		
Participants, N (%)	30 (40.5)	29 (39.2)	12 (16.2)	3 (4.1)	
Clinical diagnosis, N (%)					
CN	19 (38.8)	21 (42.9)	6 (12.2)	3 (6.1)	0.371
MCI	11 (44.0)	8 (32.0)	6 (24.0)	0	
Sex, female (%) male (%)	21 (44.7) 9 (33.3)	17 (36.2) 12 (44.4)	8 (17.0) 4 (10.9)	1 (2.1) 2 (7.4)	0.563
Age, y, mean (SD)	62.2 (7.9)	64.8 (9.7)	67.7 (8.5)	71.7 (11.0)	0.148
Education, y, mean (SD)	12.8 (4.0)	13.5 (3.3)	11.2 (4.4)	13.3 (3.2)	0.407
Family history of dementia, *n* (%)	14 (46.7)	16 (55.2)	7 (58.3)	2 (66.7)	0.843
MMSE, mean (SD)	28.4 (1.8)	28.1 (1.9)	27.2 (1.8)	28.5 (0.7)	0.274
ApoE status					
ε4 carrier, *n* (%)	8 (38.1)	5 (23.8)	8 (38.1)	0	0.005
non-carriers	22 (42.3)	24 (46.2)	3 (5.8)	3 (5.8)	

Abbreviations: AD = Alzheimer’s Disease, CN = cognitively normal, MCI = mild cognitive impairment, SD = standard deviation, MMSE = Mini Mental State Examination.

**Table 4 medicina-58-01179-t004:** Cognitively normal (CN) prospective clinical status based on biomarker category classification.

			Diagnosis Conversion		
		Total (45)	CN	MCI	Dementia
**Biomarker category**	Normal AD biomarkers	16	16 (100%)	0	0
	AD pathologic change	20	19 (95%)	1 (5%)	0
	AD disease	6	2 (33.3%)	2 (33.3%)	2 (33.3)
	Non-AD pathologic change	3	2 (66.7%)	1 (33.3%)	0

Abbreviations: CN = cognitively normal, MCI = mild cognitive impairment, AD = Alzheimer’s Disease.

**Table 5 medicina-58-01179-t005:** Mild cognitive impairment (MCI) prospective clinical status based on biomarker category classification.

			Diagnosis Conversion		
		Total (21)	CN	MCI	Dementia
**Biomarker category**	Normal AD biomarkers	9	2 (22.2%)	5 (55.6%)	2 (22.2%)
	AD pathologic change	6	1 (16.7%)	2 (33.3%)	3 (50%)
	AD disease	6	0	2 (33.3%)	4 (66.7%)
	Non-AD pathologic change	-	-	-	-

Abbreviations: CN = cognitively normal, MCI = mild cognitive impairment, AD = Alzheimer’s Disease.

## Data Availability

Not applicable.

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
