# Peer review of "Initial Data and a Clinical Diagnosis Transition for the Aiginition Longitudinal Biomarker Investigation of Neurodegeneration (ALBION) Study"

_medicina, 2022, doi:10.3390/medicina58091179_

Round 1
Reviewer 1 Report
This is a very well written progress report. As the authors acknowledge, their data collection is incomplete, their subject pool is small, and few conclusions can be derived at this stage.
One question: about 55% of cognitively normal subjects fell on the AD continuum. 56% of MCI patients fell on that continuum. You conclude "MCI patients were more likely to harbor an AD biochemical profile compared to CN." Please explain what seems to be a discrepancy.
Author Response
Thank you for the time and effort to review our manuscript.
Regarding the comment: "about 55% of cognitively normal subjects fell on the AD continuum. 56% of MCI patients fell on that continuum. You conclude "MCI patients were more likely to harbor an AD biochemical profile compared to CN." Please explain what seems to be a discrepancy."
Indeed, the overall proportion of CN or MCI participants that fell within the AD continuum was similar (55% vs 56%). As far as the “Alzheimer’s disease” biochemical profile is concerned (i.e. A+T–(N)+, A+T+(N)– or A+T+(N)+), there was a numerical overbalance in favor of the MCI group (24% vs 12%)- however, not statistically significant as per table 3 .
We have now removed the relevant statement.
Sincerely,
A.Daskalaki
Reviewer 2 Report
The strength of this publication lies in its ease in interpretation and applicability. While not novel, CSF biomarkers of AD are routine portions of clinical care now, it is important to demonstrate that these biomarkers are associated with clinical progression in diverse cohorts. In this case, a Greek cohort. While the sample size is modest (74 participants), I was encouraged that the findings are in line with previous publications showing that AD pathologic changes are associated with an increased likelihood of clinical progression. As the author's mention, a larger sample size would have produced more robust results. Overall, this study is important because it touching on biomarkers in a diverse population, but in the broader context of CSF biomarkers is relatively low impact.
I a few minor edits comments which may clarify some points.
Could you please cite the international guidelines for CSF biomarker collection that are utilized (line 155)
Could you make a table comparing participants with CSF biomarkers available to those without CSF biomarkers, I understand the line that there was no statistically significant difference between the two groups but I would like to see those values rather than the demographics of the larger cohort from which this smaller sub-population was drawn.
Please define criteria used for MCI and dementia. I appreciate that this may be based on clinical diagnosis, but was this clinical diagnosis done using Clinical Dementia Rating scale (or a similar scale) or based on neuropsychologic parameters (more than 2 standard deviations below normal in several catagories). Some clarification and references would be helpful.
Line 260 Beginning with "Probably..." is an interpretation of the results and should probably be moved to the discussion. Within the same sentence "limmitting" is a misspelling.
Author Response
We appreciate your time and effort in reweiwing our manuscript. Regarding
Comment 1:"Could you please cite the international guidelines for CSF biomarker collection that are utilized (line 155) "
Our lab protocol has been designed based on the international guidelines for CSF biomarker collection, as per following reference:
Teunissen CE, Tumani H, Engelborghs S, Mollenhauer B. Biobanking of CSF: international standardization to optimize biomarker development. Clin Biochem. 2014 Mar;47(4-5):288-92. doi: 10.1016/j.clinbiochem.2013.12.024. Epub 2014 Jan 2. PMID: 24389077. (Table 2: Collection protocol for CSF and blood pairs for biobanking).
We have now included this reference.
Comment 2: "Could you make a table comparing participants with CSF biomarkers available to those without CSF biomarkers, I understand the line that there was no statistically significant difference between the two groups but I would like to see those values rather than the demographics of the larger cohort from which this smaller sub-population was drawn."
We have now included the following table (Table 2: Participants’ characteristics with and without CSF biomarkers profile) in the updated version of the manuscript
|
|
Participants with biomarkers (n=74) |
Participants without biomarkers (n=64) |
p-value |
|
Sex, female (%) |
47 (34.1) |
44 (31.9) |
.517 |
|
Age, y, mead (SD) |
64.5 (9.0) |
63.0 (9.0) |
.329 |
|
Education, y, mean (SD) |
13.0 (3.7) |
13.6 (3.9) |
.332 |
|
MMSE, mean (SD) |
28.13 (1.81) |
28.25 (2.1) |
.717 |
|
Clinical Diagnosis CN MCI |
49 25 |
50 14 |
.121 |
Comment 3: "Please define criteria used for MCI and dementia. I appreciate that this may be based on clinical diagnosis, but was this clinical diagnosis done using Clinical Dementia Rating scale (or a similar scale) or based on neuropsychologic parameters (more than 2 standard deviations below normal in several catagories). Some clarification and references would be helpful. "
For the Definition of MCI within the cohort, we applied the Petersen criteria “a measurable deficit-below 1.5 SD- in cognition in at least one domain, in the absence of dementia or impairment in everyday functioning “, based on the following references:
Petersen RC, Smith GE, Waring SC, Ivnik RJ, Tangalos EG, Kokmen E. Mild cognitive impairment: clinical characterization and outcome. Arch Neurol. 1999 Mar;56(3):303-8. doi: 10.1001/archneur.56.3.303. Erratum in: Arch Neurol 1999 Jun;56(6):760. PMID: 10190820.
Petersen RC. Mild cognitive impairment as a diagnostic entity. J Intern Med. 2004 Sep;256(3):183-94. doi: 10.1111/j.1365-2796.2004.01388.x. PMID: 15324362.
Participants with MCI diagnosis were further classified as amnestic MCI (aMCI) or non- amnsestic MCI (naMCI), when the abnormal cognitive performance was observed in a non-memory cognitive domain (such as language, attention/speed, executive or visuospatial domain). Both aMCI and naMCI could be single or multidomain, depending on the presence of abnormal performance in other cognitive domains.
For the Definition of Dementia within the cohort, we applied the DSM-IV-TR criteria as well as the core clinical criteria for all-cause dementia of NIA-AA, as per following references:
American Psychiatric Association. Diagnostic and Statistical Manual of Mental Disorders. 4th ed ed. Washington, DC2000.
McKhann GM, Knopman DS, Chertkow H, Hyman BT, Jack CR Jr, Kawas CH, Klunk WE, Koroshetz WJ, Manly JJ, Mayeux R, Mohs RC, Morris JC, Rossor MN, Scheltens P, Carrillo MC, Thies B, Weintraub S, Phelps CH. The diagnosis of dementia due to Alzheimer's disease: recommendations from the National Institute on Aging-Alzheimer's Association workgroups on diagnostic guidelines for Alzheimer's disease. Alzheimers Dement. 2011 May;7(3):263-9. doi: 10.1016/j.jalz.2011.03.005. Epub 2011 Apr 21. PMID: 21514250; PMCID: PMC3312024.
The diagnosis of probable or possible AD was made according to the National Institute of Neurological and Communicative Disorders and Stroke/Alzheimer Disease and Related Disorders Association (NINCDS/ADRDA) criteria:
McKhann G, Drachman D, Folstein M, Katzman R, Price D, Stadlan EM. Clinical diagnosis of Alzheimer's disease: report of the NINCDS-ADRDA Work Group under the auspices of Department of Health and Human Services Task Force on Alzheimer's Disease. Neurology. 1984 Jul;34(7):939-44. doi: 10.1212/wnl.34.7.939. PMID: 6610841.
We have now included the relevant information and references.
Comment 4: Line 260 Beginning with "Probably..." is an interpretation of the results and should probably be moved to the discussion. Within the same sentence "limmitting" is a misspelling.
We have now made the above changes.
Sincerely,
A.Daskalaki